# Pivotal *Shigella* Vaccine Efficacy Trials—Study Design Considerations from a *Shigella* Vaccine Trial Design Working Group

**DOI:** 10.3390/vaccines10040489

**Published:** 2022-03-22

**Authors:** Patricia B. Pavlinac, Elizabeth T. Rogawski McQuade, James A. Platts-Mills, Karen L. Kotloff, Carolyn Deal, Birgitte K. Giersing, Richard A. Isbrucker, Gagandeep Kang, Lyou-Fu Ma, Calman A. MacLennan, Peter Patriarca, Duncan Steele, Kirsten S. Vannice

**Affiliations:** 1Departments of Global Health and Epidemiology, University of Washington, Seattle, WA 98105, USA; 2Department of Epidemiology, Emory University, Atlanta, GA 30322, USA; erogaws@emory.edu; 3Department of Medicine, Infectious Diseases and International Health, University of Virginia, Charlottesville, VA 22908, USA; jp5t@hscmail.mcc.virginia.edu; 4Department of Pediatrics, Medicine, Epidemiology, and Public Health, University of Maryland, Baltimore, MD 21201, USA; kkotloff@som.umaryland.edu; 5Enteric and Sexually Transmitted Infections Branch, National Institutes of Health, Rockvile, MD 20892, USA; cdeal@niaid.nih.gov; 6Immunization, Vaccines, and Biologicals Department, World Health Organization, 1211 Geneva, Switzerland; giersingb@who.int (B.K.G.); isbruckerr@who.int (R.A.I.); 7Department of Gastrointestinal Sciences, Christian Medical College, Vellore 632004, Tamil Nadu, India; gkang@cmcvellore.ac.in; 8Enteric and Diarrheal Diseases Program Strategy Team, Bill & Melinda Gates Foundation, Seattle, WA 98102, USA; lyou-fu.ma@gatesfoundation.org (L.-F.M.); calman.maclennan@gatesfoundation.org (C.A.M.); duncan.steele@gatesfoundation.org (D.S.); kirsten.vannice@gatesfoundation.org (K.S.V.); 9Bill & Melinda Gates Medical Research Institute, Cambridge, MA 02139, USA; ppatriarca@immuno-vax.com

**Keywords:** vaccine trial design, pediatrics, low and middle-income countries, *Shigella*

## Abstract

Vaccine candidates for *Shigella* are approaching phase 3 clinical trials in the target population of young children living in low- and middle-income countries. Key study design decisions will need to be made to maximize the success of such trials and minimize the time to licensure and implementation. We convened an ad hoc working group to identify the key aspects of trial design that would meet the regulatory requirements to achieve the desired indication of prevention of moderate or severe shigellosis due to strains included in the vaccine. The proposed primary endpoint of pivotal *Shigella* vaccine trials is the efficacy of the vaccine against the first episode of acute moderate or severe diarrhea caused by the *Shigella* strains contained within the vaccine. Moderate or severe shigellosis could be defined by a modified Vesikari score with dysentery and molecular detection of vaccine-preventable *Shigella* strains. This report summarizes the rationale and current data behind these considerations, which will evolve as new data become available and after further review and consultation by global regulators and policymakers.

## 1. Introduction

*Shigella* is a critical target for diarrheal disease prevention among children in low-resource settings. Because of the lower sensitivity of culture for *Shigella*, the application of molecular diagnostics has led to substantial increases in estimates of *Shigella* incidence in etiologic studies of childhood diarrhea in low- and middle-income countries (LMICs), with the degree of increase depending on the sensitivity of the *Shigella* culture across different study settings, specimen handling protocols, recent antibiotic use, and laboratories [1,2,3]. In the multisite Etiology, Risk Factors, and Interactions of Enteric Infections and Malnutrition and the Consequences for Child Health and Development (MAL-ED) birth cohort study, *Shigella* was the most common cause of diarrhea of any severity, with an overall incidence of 26.1 episodes per 100 child years (95% confidence interval (CI) 23.8, 29.9) and site-specific incidence ranging from 4.2 to 65.2 episodes per 100 child years [2]. In the multisite Global Enterics Multicenter Study (GEMS), *Shigella* was again the most common cause of moderate-to-severe diarrhea (MSD), attributed to about 25% of all episodes and with an incidence of 2.0 episodes of MSD in infants (95% CI 1.4, 2.6), 7.0 episodes in the second year of life (95% CI 5.0, 9.0) and 2.3 episodes in children 24–59 months of age (95% CI 1.2, 3.4) per 100 child years [1]. As in MAL-ED, site specific *Shigella* MSD incidence rates varied across sites: from 0.5 to 3.7 in infants, 2.7 to 10.5 in 12–23 month olds, and 0.6 to 7.2 in 24–59 month olds per 100 child years. Multiple other studies support this high burden, including qPCR analyses of diarrhea samples obtained from rotavirus vaccine trials in India, Bangladesh, and Niger [4,5,6]. In most of these studies, the incidence of shigellosis increased substantially in the second year of life, suggesting that the majority of shigellosis could be prevented by a vaccine introduced in older infants. One notable exception is the study from Niger, in which 60.5% of severe shigellosis occurred in infants, 37.2% in children less than 9 months of age, and 9.7% in children less than 6 months of age [5]. The Global Burden of Disease Study (utilizing GEMS data) estimated more than 200,000 annual deaths due to *Shigella* diarrhea, including more than 60,000 in children under 5 years of age, further supporting the potential public health impact of a *Shigella* vaccine [7]. There is also evidence of an association between linear growth shortfalls, a marker of chronic malnutrition that may also predict impaired cognitive development, and both *Shigella* diarrhea and subclinical infection [4,8,9]. This suggests that prevention of shigellosis may lead to both improved acute and long-term outcomes among children in high prevalence settings.

The World Health Organization (WHO) has defined the primary goal of a *Shigella* vaccine development program to develop a safe, effective, and affordable vaccine to reduce mortality and morbidity due to dysentery and diarrhea caused by *Shigella* in children under 5 years of age in LMICs [10]. Multivalent *Shigella* vaccines are critical given limited evidence of heterotypic protection, and the leading candidates aim to protect against at least four serotypes, most commonly *S. sonnei, S. flexneri* 2a, 3a and 6 [11]. As described in this Supplement, there are multiple candidates at varying stages of development.

To most efficiently advance the *Shigella* vaccine development, the manufacturing and clinical trial programs must be considered well in advance to ensure they fully evaluate the product safety and efficacy in the settings in which they will be used. Such planning also helps to minimize unnecessary delays and expenses during product development by identifying, as early as possible, any *Shigella* vaccine candidates that may not be adequately efficacious in the target population and ensuring a sound scientific rationale can be provided to support clinical development, licensure, and prequalification prior to vaccine introduction. Achieving such efficiency requires that the data generated during clinical development are scientifically sound and adequately meet the needs of both regulators and policymakers. The mandate of regulatory bodies is to ensure the safety, purity, potency, and effectiveness of authorized vaccines, while other agencies and immunization technical advisory groups provide recommendations for public health use of vaccines considering not only safety and effectiveness, but also disease burden, clinical characteristics, alternatives to vaccination, programmatic suitability of the vaccine, economic considerations, health system considerations, and others [12]. A previous WHO-led consultation of experts, policy-makers, and regulators determined that a phase 3 efficacy trial among the target population will be necessary for licensure in LMICs given the feasibility of an efficacy trial, the lack of an established correlate of protection, and the concern of generalizability of the adult controlled human infection model (CHIM) to children living in LMICs [13].

The World Health Organization’s Department of Immunization, Vaccines, and Biologicals (IVB) released a preferred product characteristics profile (PPC) outlining the desired indications, target populations, safety considerations, and endpoint targets for vaccine developers and policymakers to consider [10]. The PPC outlined the goal of at least 60% efficacy of prevention of MSD attributed to vaccine-preventable *Shigella* serotypes in children aged 6–36 months, living in *Shigella* endemic settings, with a duration of protection of at least 24 months. The PPC offers a clinical case definition of MSD of medically attended diarrhea accompanied by one or more of: dehydration, dysentery, clinician decision to hospitalize (as defined in GEMS [14]) with consideration for a potential scoring system to provide flexibility in secondary outcome definitions of severity. While the PPC specifies culture confirmation as the likely microbiologic endpoint due to its serotyping ability, the document provides a rationale for the molecular methods to be considered if serotyping is possible with such methods. The PPC recommends full protection by 12 months of age to avert *Shigella*-attributed diarrhea during the period of highest *Shigella* burden.

A *Shigella* vaccine candidate could move into evaluation in a phase 3 efficacy study as early as 2024. Thus, experts are continuing to deliberate [15,16] the key aspects of the clinical trial design that will provide the sound scientific evidence that demonstrates the efficacy of the candidate vaccines in order to support best both its submission for licensure and generate the necessary data for global and national policy decisions. To support future deliberations around *Shigella* vaccine efficacy trials, the Bill and Melinda Gates Foundation and partners, including WHO, convened an ad hoc working group who met regularly from May to November 2021 to discuss options for a vaccine efficacy trial design. The objective of the working group was to build on or adapt, based on new data, specifications outlined in the WHO PPC to make proposals for the key aspects of trial design for further review and consultation by global regulators and policymakers. This working group focused on key study design decisions, including laboratory confirmation of *Shigella*, clinical case definitions, and the primary endpoint to achieve the desired indication. The following paper describes the recommendations and considerations raised in this group (summarized in Table 1) with the intention for these recommendations to be further refined as new data become available and further consultative processes continue.

## 2. Phase 3 Efficacy Clinical Trial Design Considerations

### 2.1. Indication

The proposed indication of a *Shigella* vaccine candidate would be specific only to the strains it includes. This indication is consistent with the approach taken for the licensure of prior pneumococcal conjugate vaccines [17] as well as the pentavalent rotavirus vaccine [18]. The principal rationale for this approach is to ensure the highest likelihood of successful demonstration of efficacy in a phase 3 clinical trial, since the prevention of homotypic strains is expected to be the highest. With four species, and several serotypes and sub-serotypes within the species, and limited evidence of cross-serogroup protection [19] achieving heterotypic protection against all *Shigella* strains may be challenging [20]. Although there may be cross-protection across *Shigella flexneri* serotypes and subserotypes not contained within a vaccine, as suggested in an animal model [21] and by evidence of acquired protection following natural infection [22,23,24], the degree of protection against partially or fully heterotypic strains is not known and would be best addressed in secondary outcomes of eventual vaccine trials and/or from post-licensure studies in endemic settings.

Limiting the indication to vaccine-preventable strains also has the benefit of improving comparability across efficacy estimates in populations with a heterogeneous strain composition. However, to predict the local impact of the vaccine, an understanding of the circulating serotypes and subserotypes is needed. Because the most advanced current vaccine candidates take a multivalent approach that should provide coverage against the species, serotypes, and subserotypes that cause the majority of shigellosis in target populations [11], the focus on vaccine-preventable serotypes is most likely to achieve efficacy targets and optimize public health impact.

### 2.2. Age of Administration

The incidence of *Shigella* is highest in 12–24 month olds [1,15,25]; therefore, full vaccine protection by 12 months of age should avert the greatest burden of disease. However, the risk of severe disease, growth faltering, and case fatality is the highest in infants [5,9,26,27]. A 6 and 9 month dose (the latter of which corresponds to a measles-containing vaccine) could achieve the goal of immunizing children before 12 months of age, while providing some partial protection during the vulnerable 6–12 months of life. While there is no current Expanded Program on Immunization (EPI) visit at 6 months, the RTS,S/AS01 malaria vaccine, endorsed by the WHO in October 2021 [28], is likely to be added to some EPI schedules at 6 months, potentially opening up the 6 month timepoint for a *Shigella* vaccine in countries planning to introduce this vaccine.

The earlier introduction of a *Shigella* vaccine, such as to avert the poor prognosis following *Shigella* infection in younger infants, could also be considered. However, this would need to be weighed against the practical consideration of an already-crowded early EPI vaccination schedule and potential interaction with maternal antibodies [29]. Further, the vaccine may be more immunogenic in previously exposed and/or older children as it has been observed for the NIH *S. sonnei* O-antigen-rEPA conjugate vaccine [30]. Because *Shigella* is primarily transmitted human-to-human, there may also be substantial indirect protection by successful vaccines to children in the youngest age groups; however, this would need to be explored in future studies.

### 2.3. Clinical Case Definition

The optimal case definition for *Shigella* vaccine efficacy trials will characterize the subset of shigellosis that captures both the clinically relevant and vaccine-preventable burden of disease to optimize a trial’s ability to detect an effect and meet efficacy targets. Moderate or severe diarrhea and/or dysentery is expected to satisfy these criteria based on studies of *Shigella* vaccines in controlled human infection models (CHIMs) and precedents set by other vaccines. Higher efficacy against more severe subsets of disease has been previously observed in a recent phase 2b CHIM study of the Flexyn2a vaccine [31], for typhoid [32], and rotavirus [33]

There are many options for defining the severity of diarrhea, including with clinical scoring systems, such as the Vesikari [34], Clark [35], CODA [36], and Dhaka [37] scores, and modifications of these scores developed to fit within existing studies and/or settings [16,38,39,40]. A score developed specifically to predict 14 day mortality among *Shigella* qPCR or culture-positive cases was also developed in a post hoc analysis of the GEMS data [26]. Finally, *Shigella* CHIM studies have used a variety of clinical definitions to classify standardized endpoints, as previously summarized [15], though some of the signs are difficult to discern in young children [41].

The MSD definition used in GEMS [42], which detects medically attended diarrhea accompanied by WHO-defined signs of “some” dehydration (sunken eyes more than usual and decreased skin turgor), dysentery, or the requirement for intravenous rehydration or hospitalization, is specified in the PPC. This definition was derived by a multinational steering committee of experts in diarrheal disease and diarrheal disease research as described in detail elsewhere [43,44]. MSD would be simple to apply in a trial, as it can be classified quickly at a single timepoint (i.e., at presentation to care), and, unlike the mVesikari score, MSD does not require that characteristics for the entire duration of a diarrhea episode be captured through follow-up by the study. A main disadvantage to this definition for a trial endpoint is that, as a solely dichotomous definition, it would not lend itself to examination of various severity score cut-offs, such as “severe” or “very severe”. Additionally, it does not include other clinically relevant features of severity found to be associated with mortality, such as diarrhea duration, vomiting, and maximum number of loose stools per day [26,27]. This definition also may not be sufficiently severe as a child with diarrhea and sunken eyes only (an indication of “some” dehydration by the WHO definition which may be subjective [42]) would qualify for the primary endpoint.

Of clinical endpoint options, a moderate or severe definition based on the modified Vesikari score (mVesikari) used in rotavirus vaccine trials (described in Table 2), in addition to dysentery, may be the most appropriate for a field-based *Shigella* vaccine efficacy trial. The mVesikari score was widely used and accepted in pediatric rotavirus field efficacy trials used for vaccine licensure [33,45,46], and covers a wide range of signs and symptoms indicative of diarrhea severity. The score includes the duration of diarrhea, a risk factor for *Shigella* mortality [26], which is not captured in other definitions. Unlike a dichotomous definition, the mVesikari score also enables secondary analyses using various cut-offs, such as severe (score ≥ 11) and very severe (score ≥ 15). Finally, while the Vesikari score was originally developed to distinguish rotavirus diarrhea from other etiologies [34], it comprehensively captures signs and symptoms of watery diarrhea that are agnostic to underlying etiology. The mVesikari score has been used in a norovirus vaccine trial [47] and two recent probiotic treatment trials [48,49] with primary endpoint definitions of mean mVesikari score and mVesikari score ≥ 9. Because the Vesikari score does not capture evidence of inflammatory destruction of the intestinal mucosa, which is the hallmark of *Shigella* pathogenesis, we recommend dysentery also satisfy the case definition. The requirement that dysentery also meet the definition of diarrhea (3 or more abnormally loose or watery stools), as was required in the GEMS-MSD definition [14], will add specificity to the case definition beyond one single dysenteric stool. In the MAL-ED study, only 2.7% (n = 3/110) of dysentery episodes attributable to *Shigella* by qPCR had fewer than 3 loose stools in a 24 h period.

The primary clinical case definition of moderate or severe diarrhea (3 or more abnormally loose or watery stools and mVesikari ≥ 9 [or 7] points) or dysentery (3 or more abnormally loose or watery stools with visible blood present in ≥1 stool) would be applied retrospectively (i.e., in the data analysis phase) rather than at presentation to care to incorporate symptoms occurring throughout the diarrheal episode (as was performed in rotavirus vaccine trials). The capture of symptoms for the duration of the episode may be difficult in some settings, especially when children with diarrhea cases are not admitted to hospital and/or return home before the end of the episode. Memory recall devices, as was successfully utilized in the GEMS study, could help in these instances, although may require validity testing in settings with low adult literacy rates.

Options for the mVesikari score cut-off to capture both moderate and severe forms of diarrhea include 7, 9, and 11. Severe diarrhea by the mVesikari score ≥ 11 was used as the primary endpoint in phase 3 pediatric rotavirus field efficacy trials [33,45,46]. The thresholds for moderate diarrhea as defined by the mVesikari score vary across publications with some defining moderate as 7–10 points [50] and others as 9–10 points [38].

Vomiting, which is generally not associated with shigellosis, contributes 6 out of 20 points to the mVesikari score. Therefore, a moderate or severe definition of ≥9 (or ≥7) points may be more appropriate than ≥11 points to accommodate multiple pathways of meeting the clinical endpoint definition without vomiting. The percentage of non-dysenteric diarrhea episodes meeting the mVesikari score definitions of ≥11, ≥9, and ≥7 that were not accompanied by vomiting were 29.0%, 39.3%, and 52.0%, respectively, in the MAL-ED study. In comparison, 57.1% of non-dysenteric diarrhea episodes meeting the GEMS-MSD definition in the MAL-ED study were not accompanied by vomiting. In GEMS, the majority of *Shigella*-attributable non-dysenteric MSD diarrhea episodes meeting the moderate or severe case definitions by mVesikari of ≥11, ≥9, ≥7 and did not include excessive vomiting (32.5%, 47.6%, and 62.5%, respectively), which confirms the many pathways to meeting case definitions defined by the mVesikari score without excessive vomiting. A more inclusive cut-off, such as ≥9 and ≥7, would ensure that vomiting is not overly represented in the case definition. However, vomiting remains an important indicator of severe shigellosis, which justifies its inclusion in the severity score. In GEMS, 24% of children with *Shigella*-attributable GEMS MSD had excessive vomiting (≥3 vomiting episodes/day) and excessive vomiting was associated with 14 day mortality (age- and site-adjusted hazard ratio: 2.5, 95% CI: 1.1–5.9) [26].

### 2.4. Microbiologic Case Definition

There are two possible diagnostic approaches for the confirmation of *Shigella* diarrhea in vaccine efficacy studies, culture or quantitative polymerase chain reaction (qPCR). There is substantial precedent for the use of molecular diagnostics to identify the etiology of diarrhea. Molecular diagnostic panels are now frequently used in clinical practice in high income countries to identify the causes of diarrhea, including *Shigella*, and were endorsed as an alternative to culture in the 2017 Infectious Diseases Society of America clinical guidelines [51,52].

For a vaccine efficacy trial, there are several clear advantages to qPCR over culture. First, as the trials are expected to be multisite studies, qPCR offers more consistent test characteristics between sites. Prior studies in the target populations have revealed substantial heterogeneity in culture sensitivity between sites despite significant efforts to standardize sample collection, transport, and *Shigella* isolation. Even in best-case scenarios, e.g., a known outbreak with samples sent to clinical laboratories, the use of molecular approaches has been shown to increase shigellosis case detection [53]. All of the determinants of the relative insensitivity of culture for detection of *Shigella* are not known, though exposure to antibiotic therapy prior to stool sampling is a major factor [3], and there is evidence from target populations that the youngest children are most likely to be culture-negative despite having detectable *Shigella* at quantities associated with diarrhea [26]. Second, a substantial logistical benefit to qPCR for research settings is that it does not need to be performed in the real-time enabling inclusion of research sites that might not otherwise have a microbiology laboratory near-by.

There are two important limitations to qPCR. First, the high sensitivity requires careful interpretation of detections of low quantities of *Shigella* in diarrheal stools that may not be etiologic. For this reason, a quantitative cut-off would be needed to improve the clinical specificity of the microbiologic endpoint, especially for studies of children in low-resource settings where *Shigella* carriage in the absence of diarrhea is most common [2]. Importantly, even when this cut-off is employed, molecular detection offers a substantial increase in sensitivity over culture. Second, the most commonly used gene target, *ipaH*, is conserved between *Shigella* and enteroinvasive *E. coli* (EIEC), and thus some *ipaH* detection may represent the latter. However, because the suggested indication is vaccine-preventable shigellosis, the identification of a specific *Shigella* serotype or species would exclude the possibility of misclassifying EIEC as *Shigella* and ensure the specificity of the microbiologic endpoint. Recently, a substantial additional hurdle has been cleared, namely the identification of *S. flexneri* serotypes/subserotypes and *S. sonnei* directly from stool [54]. Thus, a culture-independent approach to detection of vaccine-preventable shigellosis is now possible.

Despite the advantages of qPCR, the inclusion of culture as a secondary endpoint remains important, as culture represents the gold standard for detection and subsequent serotyping and antibiotic resistance testing of *Shigella* isolates. To date, there are no gene targets for *Shigella dysenteriae* and *Shigella boydii*, therefore establishing protection against these species will require culture-based serotyping. Further, any detection of *Shigella* by culture would be reasonably considered etiologic in a vaccine trial without asymptomatic controls, and thus the derivation of a cut-off or other schema for the attribution of etiology would not be necessary. Powering a pivotal efficacy trial for both microbiologic endpoints could estimate efficacy against vaccine-preventable shigellosis using gold standard methods as well as address any lingering doubts about the clinical relevance and serotyping ability of qPCR, which would pave the way for subsequent trials to rely on a single diagnostic method.

### 2.5. Primary Endpoint

We suggest the primary endpoint of pivotal *Shigella* vaccine trials be the efficacy of the vaccine against the first episode of acute moderate or severe diarrhea by mVesikari and/or dysentery caused by molecularly determined *Shigella* strains contained within the vaccine. This endpoint will maximize the sensitivity of the trial’s ability to detect an effect of the vaccine by focusing on the most sensitive diagnostic tool, as well as strains (those contained within the vaccine) and diarrheal episodes (moderate to severe) most likely to be prevented by the vaccine. By limiting the endpoint to first, rather than all, vaccine-preventable moderate-to-severe *Shigella* episodes during the trial follow-up period, the trial will avoid the possibility of attenuating any effect of the vaccine due to the placebo group acquiring natural immunity after a first infection during follow-up. Pivotal rotavirus vaccine trials [33,45] similarly focused primary endpoints on first episodes, as did the pivotal malaria (RTS,S) trial [55]. Notably, the first episode of *Shigella* diarrhea during the pivotal trial may not be children’s first exposure to *Shigella*: data from the eight LMICs included in the MAL-ED suggests that approximately 10% of children are infected with *Shigella* by 6 months of age and approximately 38% by 12 months [56] and early *Shigella* exposure was also found in Niger [5]. Such vaccine priming may be a crucial determinant of vaccine efficacy and, therefore, pivotal trials will require enrollment of an adequately sized immunogenicity subset to determine efficacy estimates by baseline serostatus, though serological cross-reactivity and maternal antibody may complicate this assessment.

### 2.6. Primary Endpoint Ascertainment

In the trial, we suggest that diarrhea cases be ascertained in health facilities, where clinical information and fecal samples can be optimally obtained and standardized. All children enrolled in the vaccine trial can be encouraged to attend these health facilities during episodes of illness. Case definitions and microbiologic testing would be applied to all enrolled children presenting to health facilities with diarrhea to ensure primary and secondary endpoints can be evaluated.

### 2.7. Secondary Endpoints

Leveraging the rich trial infrastructure and data, pivotal trials will also capture several policy and field-relevant secondary endpoints. All *Shigella* moderate or severe diarrhea and/or dysentery episodes, irrespective of serotype, will be a key secondary outcome to include, enabling the assessment of cross-protection against *S. flexneri* serotypes and subserotypes not contained within the vaccine. The longevity of protection, through evaluation of not only first but also subsequent episodes, will also be critical to policymaking decisions and can be evaluated as a secondary outcome either in pivotal trials or in post-licensure studies.

Because the vaccine-preventable subset of *Shigella*-associated disease in the target population is not well defined, including multiple secondary endpoints across a broad range of severities is necessary to capture the relationship between vaccine efficacy and disease severity. Alternative definitions of disease severity, such as very severe diarrhea and diarrhea of any severity as determined by modified Vesikari as well as dysentery alone will be important to include. In exploring various cut-offs of mVesikari, it will be most interpretable if the definitions use a single cut-off (e.g., ≥7) to capture severity at or above a certain level rather than incorporating two mutually exclusive cut-offs to identify specific levels of severity (e.g., 7–11 corresponding to moderate disease). This nuance is important because, if the vaccine does not completely prevent shigellosis, but rather reduces the severity of disease, efficacy estimates against mild or moderate disease alone may be uninformative (null or even negative) due to the vaccine converting severe episodes to mild or moderate episodes. The Working Group also proposes to include the GEMS-MSD and GEMS-less severe diarrhea (LSD) definitions [57] to allow comparisons to existing observational studies. Finally, hospitalization, while included in the Vesikari and GEMS-MSD severity definitions, should also be reported separately as a severity indicator due to its clear cost implications, which will be relevant to policymakers.

As outlined in the WHO PPC and other publications [15,16], additional secondary endpoints should include change in length/height for age z-score to capture the potential benefit of a *Shigella* vaccine in averting linear growth faltering. Antibiotic use and resistance should also be captured as secondary endpoints in a vaccine trial because diarrhea is commonly treated with antibiotics in an estimated 44–74% of watery diarrhea cases and 75–94% of dysentery cases [9,58], despite syndromic indications for use being limited to dysentery and suspected cholera [59]. This frequent antibiotic use may not only lead to resistance acquisition in pathogenic bacteria, such as *Shigella*, *Salmonella*, and pathogenic *E. coli,* but also in commensal gut bacteria serving as a reservoir for future resistant gene sharing. Antibiotic resistance could therefore be evaluated in not only *Shigella* isolates, but also in easily culturable *E.coli* as a proxy for antibiotic resistance in gut flora.

### 2.8. Trial Power

Power calculations for phase 3 *Shigella* vaccine efficacy trials depend on decisions about the clinical case definition, the diagnostic method used for laboratory confirmation, the strain specificity of the primary endpoint, expected efficacy, and the duration of follow-up. Calculated sample sizes also vary based on assumptions of disease incidence that will depend, in part, on the levels of healthcare-seeking behavior observed in trial cohorts compared to the levels previously observed in observational studies. For power calculations, we based the expected incidence of the primary outcome (mVesikari ≥ 7, ≥9, or ≥11 and/or dysentery) on the observed *Shigella* MSD incidence in the GEMS study. We also assumed a quadrivalent vaccine covering *S. flexneri* 2a, 3a, 6 and *S. sonnei*. We calculated sample sizes required to achieve 90% power to detect a vaccine efficacy of 60% with 10% dropout, 1:1 group allocation, a two-sided test, and a null hypothesis of 20% (Table 3). For a primary endpoint defined by mVesikari ≥ 9 and/or dysentery with microbiologic confirmation by qPCR for the vaccine-preventable serotypes, a trial with a single year of follow-up would need to enroll 6378 children. If qPCR was only able to type 80% of molecularly confirmed *Shigella*, such as due to suboptimal sensitivity of serotyping targets compared to culture, 7922 children would need to be enrolled. A similar trial requiring microbiologic confirmation by traditional culture would require 15,224 children. The sample size would quadruple if care seeking is not adequately encouraged in the trial participants and only 25% of children with diarrhea present to care at study facilities, which are the censused community-based rates observed in GEMS [60]. Sample sizes using the GEMS-MSD definition would be almost equivalent (2% larger) to those with the mVesikari ≥ 9 and/or dysentery definition since 98% of GEMS-MSD episodes met that definition. 

Sample size calculations assuming prevention of all *Shigella* vs. vaccine-preventable *Shigella* strains are also presented in Table 3. These calculations assume an equal target efficacy (60%) regardless of serotype specificity (e.g., we assume the vaccine could achieve 60% efficacy against both vaccine-preventable serotypes/subserotypes and non-vaccine serotypes/subserotypes). If we, instead, assume 60% efficacy against vaccine-preventable serotypes, but that there is no heterotypic immunity (0% efficacy) for non-vaccine serotypes, the expected vaccine efficacy would be 39% overall based on the expected distribution of circulating serotypes. In this scenario, the total trial size required for an all-*Shigella* endpoint defined by mVesikari ≥ 9 and/or dysentery and qPCR with 1 year follow-up would increase from n = 4004 to n = 20,684, which is more than three times larger than the size required for the corresponding endpoint that is specific to the vaccine-preventable serotypes (n = 6378).

Some regulatory bodies may specify a higher or lower than 20% limit on the lower bound of 95% confidence interval for vaccine efficacy estimates. These limits, or equivalent changes to the null hypothesis, change the required sample size substantially. For a primary endpoint defined by mVesikari ≥ 9 and/or dysentery with microbiologic confirmation by qPCR for the vaccine-preventable strains, a trial with a single year of follow-up would need to enroll 9841 children when limiting the lower 95% confidence interval limit to 30% (vs. 6378 with a 20% lower bound), and 23,485 children for the same clinical case definition, but with microbiologic confirmation by culture. A 10% lower limit would require 4649 children be enrolled, or 11,101 with culture confirmation.

## 3. Conclusions

A phase 3 field trial powered to determine the efficacy of a *Shigella* vaccine against the first episode of acute moderate or severe diarrhea and/or dysentery caused by molecularly determined *Shigella* strains contained within the vaccine is feasible. The primary endpoint proposed in this paper utilizes the best available technology for sensitive assays and a clinical case definition that captures moderate and severe disease, but also provides flexibility to capture more mild cases to characterize better the full public health value of a *Shigella* vaccine. A carefully designed licensure trial will ensure regulators and policymakers can make evidence-based recommendations. This paper reflects the beginning of a broader consultative process, to be led by WHO, to collect additional inputs into efficacy trial design to support these ultimate objectives. By proactively trying to outline the design aspects before a product is ready to enter an efficacy trial, it is hoped that timelines for starting the trial will be reduced. At the same time, it is anticipated that the trial design characteristics described in this paper will be further refined as additional inputs are obtained and new data become available from other relevant studies.

## Figures and Tables

**Table 1 vaccines-10-00489-t001:** Key considerations for the design of pivotal *Shigella* vaccine efficacy trials.

Primary endpoints should focus on the intended serotype/subserotype targets included in the vaccine and secondary outcomes should assess the protection against additional *Shigella* strains. The optimal timing of vaccination will need to consider the number of doses and dosing intervals as well as the burden of disease, including in the 6–12 month age group. A clinical endpoint of moderate or severe diarrhea defined by a modified Vesikari score (mVesikari) and/or dysentery provides maximal flexibility to evaluate efficacy against a range of clinical severities. Molecular confirmation of clinically relevant shigellosis is more sensitive than traditional stool culture and thus maximizes study power as a primary endpoint. Stool cultures should also be employed and powering the trial with culture-confirmed endpoints could be considered to maximize cross-study comparisons and interpretability. To sufficiently characterize the vaccine performance in both participants with and without evidence of prior *Shigella* exposure (i.e., pre-existing antibodies), an immunogenicity subset of sufficient size should be enrolled.

**Table 2 vaccines-10-00489-t002:** Components of the modified Vesikari score used in the rotavirus vaccine trials.

Score Component	Modified Vesikari Score
Duration of diarrhea	1–4 days (1 point) 5 days (2 points) ≥6 days (3 points)
Max number of stool in 24 h period	1–3 diarrheal stools (1 point) 4–5 diarrheal stools (2 points) ≥6 diarrheal stools (3 points)
Duration of vomiting	1 day (1 point) 2 days (2 points) ≥3 days (3 points)
Max number of vomiting episodes in 24 h period	1 (1 point) 2–4 (2 points) ≥5 (3 points)
Maximum recorded temperature	Rectally: 37.1–38.4 °C (1 point) 38.5–38.9 °C (2 points) ≥39.0 °C (3 points) Axillary: 36.6–37.9 °C (1 point) 38.0–38.4 °C (2 points) ≥38.5 °C (3 points)
Dehydration (based on WHO-defined dehydration categories)	None (0 points) Some (2 points) Severe (3 points)
Treatment	None (0 points) Rehydration (1 point) Hospitalization (2 points)
Score categories	Mild illness (0–6 points) Moderate illness (7–10 points) Severe illness (≥11 points) [Total out of 20 points]

**Table 3 vaccines-10-00489-t003:** Preliminary sample size calculations targeting efficacy with a lower 95% confidence interval bound of 20% for pediatric efficacy trials of *Shigella* vaccines based on data from the GEMS study.

Case Definition	Diagnostic Specificity of Primary Endpoint	Expected Incidence from GEMS (per 100 Child Years)	Total Trial Size Required ^§^			
			Assuming 100% Medical Attendance		Assuming 25% Medical Attendance ^#^	
			2 Years of Follow-Up	1 Year of Follow-Up	2 Years of Follow-Up	1 Year of Follow-Up
Vesikari ≥ 11 or dysentery	All *Shigella*; culture	1.3 *	5973	12,056	24,224	48,556
	All *Shigella*; qPCR	3.1 ^†^	2413	4935	9979	20,068
	Vaccine-preventable (VP) *Shigella*; culture	0.8 ^‡^	9291	18,693	37,497	75,101
	VP *Shigella*; qPCR	2.0 ^‡^	3868	7845	15,800	31,711
	VP *Shigella*; qPCR, assuming only 80% typable	1.6 ^‡,§§^	4814	9739	19,589	39,288
Vesikari ≥ 9 or dysentery	All *Shigella*; culture	1.6 **	4851	9812	19,732	39,576
	All *Shigella*; qPCR	3.8 ^††^	1947	4004	8118	16,346
	VP *Shigella*; culture	1.0 ^‡^	7557	15,224	30,556	61,222
	VP *Shigella*; qPCR	2.4 ^‡^	3133	6378	12,863	25,839
	VP *Shigella*; qPCR, assuming only 80% typable	2.0 ^‡,§§^	3905	7922	15,952	32,017
Vesikari ≥ 7 or dysentery	All *Shigella*; culture	1.7 *^,^**	4607	9326	18,762	37,631
	All *Shigella*; qPCR	4.0 ^†,††^	1846	3802	7715	15,539
	VP *Shigella*; culture	1.1 ^‡^	7181	14,472	29,053	58,219
	VP *Shigella*; qPCR	2.5 ^‡^	2974	6059	12,228	24,567
	VP *Shigella*; qPCR, assuming only 80% typable	2.1 ^‡,§§^	3709	7528	15,167	30,444

^§^ Assuming vaccine efficacy of 60%, 90% power, 10% dropout, 1:1 group allocation, 2-sided test. ^#^ Incidence was multiplied by 0.25, the average proportion of episodes that sought care as reported previously [9]. Care-seeking proportion is likely to be highly conservative, since not all facilities were surveilled, and a vaccine trial will encourage healthcare seeking at trial facilities. * Weighted average of age-specific *Shigella* incidence by culture in GEMS [25] multiplied by the proportion of MSD cases in GEMS expected to meet the proposed definition by Vesikari ≥ 11 and/or dysentery (76%). Note: this is conservative since some non-MSD episodes will also meet the Vesikari-based definition ≥ 11 (~5% estimated from MAL-ED). ^†^ Weighted average of age-specific *Shigella* incidence by qPCR in GEMS [1] multiplied by the proportion of MSD cases in GEMS expected to meet the proposed definition by Vesikari ≥ 11 + dysentery (76%). ^‡^ Vaccine-preventable *Shigella* assumed to be *S. flexneri* 2a, 3a, 6 and *S. sonnei.* Vaccine-preventable incidence rates calculated by multiplying the all-*Shigella* estimates by the proportion of cases that were *S. flexneri* 2a, 3a, 6 and *S. sonnei* in GEMS (64.3% [11]). ^§§^ Assumes only 80% of all qPCR-positive vaccine-preventable *Shigella*-attributable diarrhea episodes will have valid typing results by qPCR. ** Weighted average of age-specific *Shigella* incidence by culture in GEMS [25] multiplied by the proportion of MSD cases in GEMS expected to meet the proposed definition of severe by Vesikari ≥ 9 and/or dysentery (93.2%). Note: this is conservative since some non-MSD episodes will also meet the Vesikari-based definition ≥ 9 (~13% estimated from MAL-ED). ^††^ Weighted average of age-specific *Shigella* incidence by qPCR in GEMS [1] multiplied by the proportion of MSD cases in GEMS expected to meet the proposed definition by Vesikari ≥ 9 + dysentery (93.2%).

## Data Availability

Not applicable.

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
