# Peer review of "Pivotal Shigella Vaccine Efficacy Trials—Study Design Considerations from a Shigella Vaccine Trial Design Working Group"

_vaccines, 2022, doi:10.3390/vaccines10040489_

Round 1
Reviewer 1 Report
An excellent piece of work
Shigella causes huge burden in terms of morbidity and infections lead to growth faltering and precipitate malnutrition
Only after the availability of GEMs data and finding that culture alone fails to detect nearly 33 percent of cases , the problem caused by Shigella has come to the notice However designing a vaccine is one thing and and measuring efficacy in phase 3 trials is another issue.
A major challenge is that immunity may be serotype specific So in this paper authors have very nicely summarized key design issues :
first of all how to define moderate to severe diarrhoea,then need for molecular tests, the number of patients that will be needed, and the primary end points needed for such trials.
The authors have rationalised their viewpoints and cited all relevant literature
The topic is very relevant and original .
I agree to their conclusions and consider this article very timely
Author Response
Thank you very much for your summary and feedback. We agree with the reviewer regarding the challenges in designing an efficacy trial for a Shigella vaccine targeting children in low and middle income countries and are hopeful this document serves as a launching point for continued optimization and thinking around the design of such trials.

Reviewer 2 Report
The manuscript by Pavlinac and coworkers details a study of vaccine candidates for Shigella that are approaching phase 3 clinical trials. The manuscript tries to encompass all the considerations to reach phase 3 and gives data for the working group. The work is solid and merits publication. However, add a picture or two of shigella/or the effects (this to make it more clear) and add a paragraph before the conclusion on what are the steps and requirements to go to phase III clinical trials as stated by the CDC or the agency in charge.
Author Response
Reviewer #2
- The manuscript by Pavlinac and coworkers details a study of vaccine candidates for Shigella that are approaching phase 3 clinical trials. The manuscript tries to encompass all the considerations to reach phase 3 and gives data for the working group. The work is solid and merits publication.
Response: Thank you for your feedback.
- However, add a picture or two of shigella/or the effects (this to make it more clear)
Response: Thank you for this suggestion, and we acknowledge that figures can be helpful to include. Because study design is the focus of this manuscript and we are not presenting new data on Shigella and its effects, we have not included a picture.
- and add a paragraph before the conclusion on what are the steps and requirements to go to phase III clinical trials as stated by the CDC or the agency in charge.
Response: Thank you for this suggestion. To the best of our knowledge, there are no standardized criteria set forth by the CDC and/or FDA outlining requirements to go into Phase III clinical trials. We did identify broad criteria, as outlined by the FDA (https://www.fda.gov/media/151716/download), including that the dosage has been established and that the product is reasonably immunogenic and safe (as established in preclinical and phase 1-2 trials). We have included this citation in the introduction however have not specifically called out the FDA and/or CDC because it is unlikely these bodies will be the regulatory bodies for a Shigella vaccine targeting children in LMICs.
Reviewer 3 Report
Pavlinac et al present concepts towards a Phase 3 study design for Shigella vaccines as a result of an ad hoc convening on the topic. The paper is well written, adequately describes the problem and outlines potential methods to effectively evaluate the potential vaccine candidates. Beyond minor typographical issues likely to be corrected during final editing, the only minor concern pertains to the sample size calculations presented in the paper.
It is quite possible that a regulatory agency would not accept as sufficiently demonstrative of vaccine efficacy a lower bound of the 95% confidence interval for vaccine efficacy of 0% (equivalent to the null hypothesis of no difference as described). Instead, many regulators (in particular the FDA) frequently request a lower bound of at least 30% or higher.
Given the potential pathways for licensure, experiences with regulators from target countries should be consulted to ensure these sample size calculations will support licensure. If the US FDA was to be a regulatory body reviewing the plan, there may be a need to adjust as outlined above.
Author Response
Reviewer #3
Pavlinac et al present concepts towards a Phase 3 study design for Shigella vaccines as a result of an ad hoc convening on the topic. The paper is well written, adequately describes the problem and outlines potential methods to effectively evaluate the potential vaccine candidates. Beyond minor typographical issues likely to be corrected during final editing, the only minor concern pertains to the sample size calculations presented in the paper.
Response: We appreciate your kind review and will ensure all typographical errors are fixed before print.
It is quite possible that a regulatory agency would not accept as sufficiently demonstrative of vaccine efficacy a lower bound of the 95% confidence interval for vaccine efficacy of 0% (equivalent to the null hypothesis of no difference as described). Instead, many regulators (in particular the FDA) frequently request a lower bound of at least 30% or higher. Given the potential pathways for licensure, experiences with regulators from target countries should be consulted to ensure these sample size calculations will support licensure. If the US FDA was to be a regulatory body reviewing the plan, there may be a need to adjust as outlined above.
Response: This feedback is very helpful, thank you. We have updated the sample size calculations in Table 3 assuming a lower bound of 20% and report sample size calculations for the recommended primary endpoint using lower bounds of 10% and 30% to demonstrate a range of required sample sizes. We agree that this number will be determined by the eventual regulatory body reviewing this plan.